# Carotenoid Accumulation in the *Rhododendron chrysanthum* Is Mediated by Abscisic Acid Production Driven by UV-B Stress

**DOI:** 10.3390/plants13081062

**Published:** 2024-04-09

**Authors:** Fushuai Gong, Xiangru Zhou, Wang Yu, Hongwei Xu, Xiaofu Zhou

**Affiliations:** Jilin Provincial Key Laboratory of Plant Resource Science and Green Production, Jilin Normal University, Siping 136000, China

**Keywords:** carotenoids, UV-B, *Rhododendron chrysanthum*, transcriptomics, metabolomics, ABA

## Abstract

*Rhododendron chrysanthum* (*R. chrysanthum*) development is hampered by UV-B sunlight because it damages the photosynthetic system and encourages the buildup of carotenoids. Nevertheless, it is still unclear how *R. chrysanthum* repairs the photosynthetic system to encourage the formation of carotenoid pigments. The carotenoid and abscisic acid (ABA) concentrations of the *R. chrysanthum* were ascertained in this investigation. Following UV-B stress, the level of carotenoids was markedly increased, and there was a strong correlation between carotenoids and ABA. The modifications of *R. chrysanthum*’s OJIP transient curves were examined in order to verify the regulatory effect of ABA on carotenoid accumulation. It was discovered that external application of ABA lessened the degree of damage on the donor side and lessened the damage caused by UV-B stress on *R. chrysanthum*. Additionally, integrated metabolomics and transcriptomics were used to examine the changes in differentially expressed genes (DEGs) and differential metabolites (DMs) in *R. chrysanthum* in order to have a better understanding of the role that ABA plays in carotenoid accumulation. The findings indicated that the majority of DEGs were connected to carotenoid accumulation and ABA signaling sensing. To sum up, we proposed a method for *R. chrysanthum* carotenoid accumulation. UV-B stress activates ABA production, which then interacts with transcription factors to limit photosynthesis and accumulate carotenoids, such as MYB-enhanced carotenoid biosynthesis. This study showed that *R. chrysanthum*’s damage from UV-B exposure was lessened by carotenoid accumulation, and it also offered helpful suggestions for raising the carotenoid content of plants.

## 1. Introduction

Reduced photosynthetic efficiency and membrane fluidity, together with elevated reactive oxygen species (ROS) levels, are indicative of the physiological and biochemical processes that plants often undergo under UV-B stress [1]. One of the physiological processes most vulnerable to UV-B stress is photosynthesis, which is the primary pathway of the plant matter cycle. The photosystem is also highly delicate and prone to damage under stressful environments like high light levels and low temperature [2]. The chlorophyll fluorescence parameters of living plants alter in response to unfavorable environmental conditions. By examining these parameters, one can learn more about the physiological state of the plant’s processes for absorbing, converting, and transferring light energy. Plant photosynthesis is impacted by UV-B stress in a variety of ways, with one of the most significant effects being on photosynthetic pigments [3]. It affects photosynthetic pigments in two ways: first, it lowers the amount of chlorophyll and raises the amount of carotenoids in plant leaves; second, it changes the pigments’ capacity to absorb light. Research has demonstrated that, in response to short-term UV-B stress, the content and composition ratio of photosynthetic pigments in plant leaves did not change significantly. However, there was a noticeable decrease in the effective photochemical efficiency (Fv’/Fm’) and actual photochemical efficiency (ΦPSII) of leaf photosystemII (PSII), as well as similar changes in the photochemical quenching coefficients (qP) and non-photochemical quenching coefficients (NPQ) of leaf PSII [4]. Carotenoids’ more noticeable alterations than chlorophyll’s indicate that the production of carotenoids serves as a defense against environmental stresses.

Carotenoids are a class of naturally occurring pigments that function as photoreceptors and are found in several phytoplasmas of higher plants. They are crucial for effective photosynthesis and are involved in light damage [5,6]. Carotenoids serve as precursors of phytohormones, auxiliary pigments for photosynthetic processes, and scavengers of free radicals in plants [7]. Additional derivatives of carotenoid compounds are thought to function as signaling regulators in the development of plants. Furthermore, carotenoids protect plants from abiotic challenges by scavenging unilinear oxygen species, which reduces overstress [8]. It appears that sweet potatoes with overexpressed lycopene β-cyclase had much higher carotenoid content, which improved salt tolerance. Nevertheless, it has been noted that low temperature stress reduces carotenoid content. Lower β-carotene content was observed in two tea kinds when the temperature was lowered [9]. Consequently, researching the accumulation of carotenoids in UV-B-stressed *R. chrysanthum* may aid in reducing stress and promoting improved growth in the plant.

Research has been conducted on the molecular pathways via which *R. chrysanthum* withstands adversity stress. For instance, *R. chrysanthum* plants are shielded against the harmful effects of UV-B rays by the buildup of flavonoids, organic acids, amino acids, and fatty acids, as well as by improved TCA cycling and arginine production [10]. It is suggested that ABA, the MAPK cascade, and Ca^2+^ signaling form an integrated regulatory network that works in concert to regulate cold stress in *R. chrysanthum* [11]. This information will help clarify the molecular processes behind plant cold tolerance. It is not well understood, therefore, how the *R. chrysanthum* modulates signaling pathways and, in turn, controls metabolite alterations via modifications in photosynthesis.

Many phytohormones are involved in UV-B stress, most notably abscisic acid (ABA), which is widely known to be a stress hormone. Phytohormones control signaling in response to stress, as multiple studies have demonstrated [12]. Abscisic acid (ABA) is necessary to cause stomatal closure, seed dormancy, and plant responses to abiotic stressors [13]. Recognized as ABA receptors, pyrabactin resistance 1 *(PYR)/PYR1-like (PYL)* proteins bind to ABA and activate complicated signaling pathways downstream of ABA, preventing *PP2C* from inhibiting *SnRK2* [14,15,16]. There is a correlation between elevated ABA levels during cold stress and enhanced ABA production in *Oryza sativa* L. and *Arabidopsis* [17,18]. Foliar anthocyanin accumulation was promoted by ABA-driven H_2_O_2_ generation, and this improved the abiotic stress tolerance of *abr* mutants [19].

This study monitored changes in the physiology and gene and metabolite levels of *R. chrysanthum* under various treatments in order to better understand how phytohormones associated with the plant’s resistance to UV-B stress regulate changes in metabolite networks. It found that the plant could withstand UV-B stress-induced damage to its photosynthesis system by increasing its carotenoids content and that the biosynthesis of carotenoids was related to the ABA content. Further research revealed that carotenoid accumulation in *R. chrysanthum* is a plant response to UV-B stress and that ABA is a critical component influencing carotenoid accumulation in *R. chrysanthum*. Additionally, we examined the differential metabolites and genes in leaves treated with externally administered ABA and UV-B stress following UV-B stress treatment. The relationship between externally applied ABA and carotenoid production in reducing UV-B stress in the *R. chrysanthum* was described based on the facts that were available.

## 2. Results

### 2.1. Chlorophyll Fluorescence Imaging and Alterations in the Kinetic Curve in R. chrysanthum Exposed to UV-B Radiation

Chlorophyll fluorescence parameters were measured both before and after the stress in order to examine how UV-B stress affected the chlorophyll fluorescence images of *R. chrysanthum*. Fm, Fm’, Y(II), and qP images in PSII of *R. chrysanthum* treated with UV-B stress are displayed in Figure 1a. The maximum value of fluorescence, or Fm, is the intensity of fluorescence when the plant’s light reaction center is turned off in the dark-acclimated condition. A decrease in Fm indicates that the plant is in a stress-resistant state. Photochemical quenching coefficient, or qP for short, with qP greater values show that the plant uses more light energy for photosynthesis. As shown in Figure 1a, the qP in the UV-B stress-treated group showed a significant trend of decreasing after receiving antihypertensive treatment compared with that in the control group. Figure 1a shows a summary of the study’s findings. Furthermore, the UV-B stress-treated group exhibited significantly lower values for Fv/Fm, Y(II), Fm’, and Fv’/Fo’ (Figure 1b–e). These findings suggest that UV-B stress reduced *R. chrysanthum*’s maximum photochemical efficiency and decreased its light energy utilization, which in turn impaired the PSII photosystem’s potential activity and ultimately inhibited the flow of photochemical electrons into the carbon-reducing process, thereby reducing the cowpea’s photosynthetic capacity. *R. chrysanthum*’s photosynthetic capacity dramatically reduced during UV-B exposure, as seen in Figure 1f,g. This led to photoinhibition, which reduced NPQ out of self-protection and lessened photo-oxidation by dissipating the unavailable light energy as heat energy. On the other hand, ETR did not significantly alter following UV-B stress.

### 2.2. R. chrysanthum’s Carotenoid Concentration and Its Relationship with ABA

Figure 2a shows the chlorophyll a (Chl a), chlorophyll b (Chl b), and chlorophyll a/b (Chl a/b) contents of the control group and UV-B stress-treated group. The amounts of Chl a, Chl b, and Chl a/b were discovered to be lower; nonetheless, we think that the *R. chrysanthum*’s chlorophyll contents were relatively unchanged, as it withstood the immediate UV-B stress. Figure 2b shows that the amount of carotenoid was significantly greater, and the carotenoid content was crucial to *R. chrysanthum*’s resistance to UV-B irradiation. Carotenoids functioned as co-pigments in photosynthesis, absorbing the light waves that chlorophyll was unable to absorb. They then transferred the light energy from the absorbed light energy to chlorophyll, which was then used in photosynthesis. It absorbs light energy, changes it into chlorophyll, and uses the lutein cycle to control the antenna transmission capacity. This process transforms surplus excitation energy into a form that shields chlorophyll from harm. The quantity of ABA is depicted in Figure 2c, and a correlation coefficient analysis revealed a strong relationship between the carotenoid content and ABA (Figure 2d). Our hypothesis was that giving plants ABA would make them more resistant to UV-B stress.

### 2.3. OJIP Transient Curve Modifications following External ABA Application

The OJIP transient curves between the O-phase and P-phase are shown in Figure 3a. The J phase was raised with rapid UV-B therapy, as indicated in the figure, and it gradually dropped as the duration of UV-B treatment increased. It was also discovered that the I and P phases decreased. Particularly during phases I and P, the externally administered ABA therapy was shown to counteract the alterations in the OJIP curves for UV-B stress. In comparison with the stressed group, the I-P phase had a larger amplitude.

To evaluate the course of the OJIP curves for each phase of the process observed in the O-J, J-I, and I-P phases, the fluorescence data were normalized and shown as the relative variable fluorescence kinetics at each time W_O-P_ = (Ft − Fo)/(Fm − Fo) (Figure 3b). Figure 3b demonstrates that neither stress nor externally applied ABA treatments significantly altered the PF intensity at the J-I and I-P phases. In contrast, the UV-B stress-treated group showed a greater rise in PF intensity at the O-J stage when compared to the control group, which was further boosted by ABA administration.

K-band presence at 300 s is a particular marker for the lateral photoinhibition of the PSII receptor and indicates the breakdown or inactivation of the oxygenation complex (OEC). By normalizing the relative fluorescence between the O and J phases (Figure 3c), one may observe the K band. Similarly, Figure 3d calculates the ratio FK amplitude FJ − Fo (W_K_) of variable fluorescence. The K band and W_K_ were substantially higher with UV-B stress compared to the control group; however, following the UV-B stress treatments, exogenous application of ABA reduced this increase (Figure 3e). Our hypothesis is that ABA lessens the extent of damage to the donor side, which helps plants withstand UV-B stress.

Here, we observed that after UV-B stress in *R. chrysanthum*, the receptor-side related indices like Sm, φEo, and ETo/RC were dramatically changed. In particular, following stress, Sm and ETo/RC dramatically dropped, and when ABA treatment was administered, this shift started to be mitigated (Figure 3f–h).

### 2.4. JIP Test Parameters

Figure 4a shows spider plots representing the values of the JIP test parameters related to damage to both the acceptor and donor sides of PSII. While φDo, Fo, Sm, and V_J_ were higher than those of the control, ψo, φEo, and Fm were lower in *R. chrysanthum* which had been subjected to UV-B stress. ABA treatment, on the other hand, resulted in higher levels of ψ_o_, φEo, and Fm and lower values of φDo, Fo, Sm, and V_J_ in *R. chrysanthum* under UV-B stress. *R. chrysanthum* under UV-B stress showed significantly higher levels of Fv/Fm, DIo/RC, and PI_ABS_ as compared to the control treatment, as illustrated in Figure 4b–d. However, Fv/Fm and PI_ABS_ significantly decreased under UV-B stress. When *R. chrysanthum* was exposed to UV-B stress, the application of exogenous ABA dramatically raised the values of PI_ABS_ and Fv/Fm and lowered (*p* < 0.05) the values of DIo/RC.

### 2.5. Finding the Main Pathways for the Differential Metabolite Enrichment of R. chrysanthum under UV-B Stress versus UV-B Stress When ABA Is Applied Externally

In total, 17% of the metabolites were classified as terpenes (Figure 5a). The DMs from UV-B stress and externally applied ABA treatments were screened for (Appendix A) and Kyoto Encyclopedia of Genes and Genomes (KEGG) enrichment analysis was performed. The results indicate that the different metabolites were enriched in carotenoid biosynthesis and phytohormone signaling (Figure 5b). This suggests that changes in the substances involved in these pathways helped the *R. chrysanthum* defend itself against external stresses, which in turn affected the changes in related pathways. Similarly, it also suggests that ABA application influences the changes in the two metabolic pathways. Key differential metabolites in the enriched pathways are presented as a heatmap in Figure 5c.

### 2.6. Differential Genes’ KEGG Enrichment Analysis and Critical Transcription Factor Family Searches

To create a gene annotation Venn diagram, the discovered genes were annotated into KEGG, gene ontology (GO), and transcription factors (TFs) (Figure 6a). Simultaneously annotated to KEGG, GO, and TFs, we identified 909 genes. We then screened the 909 genes for differential genes (Appendix A) and examined the differential genes for KEGG enrichment. The annotated pathways, which included phytohormone signaling, plant–pathogen interactions, and interconversions of pentoses and glucuronides, are illustrated in Figure 6b. In particular, research on the potential roles of DEGs has revealed that genes with UV-B stress treatment and the post-stress application of the ABA treatment group are primarily involved in phytohormone signaling, plant circadian rhythms, and the plant MAPK signaling pathways. These pathways are indicated in red in Figure 6c. 

Important molecular switches that control how cells grow and develop in response to different stimuli are TFs. In *R. chrysanthum*, a total of 1328 TFs with varying expression abundances were found; these were grouped into 55 families based on the PlantTFDB database. Of these, there were 62 members in the WRKY family and 82 in the bHLH family. In comparison, the MYB family consisted of 154 individuals. Among others, the AP2/ERF-ERF and NAC had 93 and 55, respectively. By grouping the top 5 transcription factor family differential genes, it was possible to observe that some TF families—such as CH3, AP2-EREBP, MYB, and bHLH—were significantly up-regulated following UV-B stress treatment and continued to be so after applying ABA treatments. Conversely, other TF families—such as CH3, AP2-EREBP, MYB, and bHLH families—were significantly down-regulated following UV-B stress treatment and continued to be so after applying ABA treatment. These findings suggest that the four families in *R. chrysanthum* regulate everything from ABA signaling to carotenoid synthesis (Figure 6c). Information on the first 15 genes with *p* ≤ 0.05 is displayed in Table 1 (the remaining genes are displayed in Appendix A).

### 2.7. The Use of ABA Therapy Exogenously Encourages the Accumulation of Carotenoids

We looked into the characterization of DMs and DEGs involved in ABA biosynthesis and signal transduction in order to analyze the possible regulatory mechanisms of exogenously applied ABA on carotenoid production under UV-B stress. *CrtZ*, *LUT5*, and *ZEP*, important genes in ABA biosynthesis, were up-regulated in response to ABA treatment (Figure 7a); in contrast, *NCED* and *AAO3* were down-regulated following ABA treatment. Following ABA treatment, there was a considerable up-regulation of key ABA signaling genes such as *SnRK2*, *PYR*/*PYL*, and *AOG*, and a significant down-regulation of ABF and *PP2C*.

DMs and DEGs involved in carotenoid metabolism by UV-B stress as well as by exogenous application of ABA after UV-B stress were compared in order to better understand the relationship between ABA signaling and carotenoid biosynthesis. These key metabolites as well as the key genes were presented using schematic diagrams and heatmap insets (Figure 7a). Glycerone phosphate and phosphatidate were shown to be important metabolites associated with glycolytic pathways, the formation of the terpenoid skeleton, and carotenoid biosynthesis. These metabolites were found to be greatly reduced following UV-B stress and to be similarly reduced following the application of ABA. *TPI*, *gapN*, *PGAM*, *ENO*, *PDHA*, and *PDHB* are important glycolytic pathway genes that were found to be up-regulated both during UV-B stress and following the application of ABA under UV-B stress. The MEP pathway’s *dxr* was identified as the primary rate-limiting enzyme in the biosynthesis of terpene skeletons, and its expression demonstrated a noteworthy increase in the pathway. Additionally, the MVA pathway’s HMGR family was the most significant rate-limiting enzyme, and its change in the UV-B pathway suggested that it plays a significant role in the specific accumulation of terpene compounds. The most prominent rate-limiting enzyme in the MVA pathway, the HMGR family, did not significantly alter following UV-B stress. These findings led us to speculate that the MEP pathway might be the primary biosynthetic route for terpenoid components in *R. chrysanthum*. The expression level of *PDS* in the carotenoid biopathway is influenced by *PSY*, the rate-limiting step in carotenoid biosynthesis. In this case, *PDS* showed a significant increase following UV-B stress and an additional increase following the application of external ABA. *ZDS* also showed a similar trend of change. By combining the alterations of important metabolites and genes in the aforementioned pathways, we discovered that UV-B stress had an impact on the *R. chrysanthum*’s carotenoid biosynthesis. Additionally, we observed that externally applied ABA further impacted the carotenoid biosynthesis by influencing both the ABA biosynthesis and the signaling pathway. Furthermore, we confirmed that ABA was helpful in mitigating the harm caused by UV-B stress to the *R. chrysanthum*.

We examined DEGs of circadian rhythms and photosynthetic pathways in the *R. chrysanthum,* since light is crucial for carotenoid production (Figure 7b). The single gene *PIF3*, which codes for the photosensitive pigment A (*PHYA*), was significantly down-regulated following UV-B treatment, exhibiting a similar pattern following ABA application. Additionally, there was a drop in *CSNK2A*, the primary regulator of the plant’s circadian rhythms. Following UV-B treatment, there was a decrease in the expression of *LHCA2*, *LHCA3*, and *LHCA5* transcripts in the photosynthesis-antenna protein pathway. Expression levels decreased following the administration of ABA. Similar to the findings of Figure 3e, the photosynthetic pathway’s *psbP*, which describes the OEC-related genes on the receptor side, was markedly elevated, suggesting that OEC was compromised. The findings supported the conclusion of Figure 1 by demonstrating that the genes *petH* and *petJ*, which are involved in photosynthetic electron transport, were also dramatically changed under UV-B exposure, this suggests that PSII electron transport was compromised.

### 2.8. Analysis of DMs and DEGs Combined under UV-B Stress

DEGs and DMs with Pearson correlation coefficients (PCC) >0.8 between the control (M) and UV-B stress-treated group (N) and between the UV-B stress-treated (N) and exogenous ABA-treated group (Q) were used to create nine-quadrant plots (Figure 8a,b). Quadrant 5 has the genes and metabolites that do not differ from each other in the plot; quadrants 3 and 7 contain the genes and metabolites that have positive correlations; and quadrants 1 and 9 contain the genes and metabolites that have negative correlations. Quadrant 2 has up-regulated metabolites related to unaltered genes; quadrant 4 contains altered metabolites coupled to down-regulated genes; and quadrant 6 contains unchanged metabolites coupled to up-regulated genes. Lastly, quadrant 8 contains down-regulated metabolites together with unchanged genes. The bulk of the DEGs in both groups—26,587 DEGs after UV-B stress treatment and 2353 DEGs after exogenous ABA treatment—are found in quadrant 2. Quadrant 3 has the majority of DMs in both groups, though—14,852 DMs after UV-B stress treatment and 350 DMs after exogenous ABA treatment. Remarkably, the homologous genes may influence DMs in quadrants 3 and 7. With respect to the terpenoids found after UV-B stress treatment, 67 were found in quadrant 3 (which included 7-Hydroxycadalene, 10-Dehydrogeniposide, Adoxosidic acid, and others), and 7 were found in Quadrant 7 (which included Atractyloside G, 1-hydroxy-β-cyperone, Cinncassiol A, and others). In total, 52 flavonoids were detected in quadrant 7 after exogenous ABA treatment (Costal, 1,4-Peroxyaurol-ene, 19-Hydroxyursolic acid, etc.) and 21 terpenoids were identified in quadrant 3 (Gardenoside, Jasminoside B, Suavioside F, etc.). Among them, quadrant 3 had Adoxosidic acid, cichorioside B, Negundoin A*, and Vitexilactone* after UV-B stress treatment and after exogenous ABA treatment, while quadrant 7 contained Kankanoside E after UV-B stress treatment and after exogenous ABA treatment. As seen in Figure 8c,d, we counted the quadrant DEGs and DMs after UV-B stress treatment and after exogenous ABA treatment. In quadrants 3 and 7 (Figure 8e,f), we conducted a two-group inter-academic chordal analysis of DEGs and DMs. We discovered substantial correlations between DEGs and DMs in both the UV-B stress treatment and exogenous ABA treatment groups.

## 3. Discussion

*R. chrysanthum* has a stress response in response to UV-B radiation, which specifically affects changes in the quantity of proteins and secondary metabolites involved in photosynthesis. One of the parts of the cell that is most vulnerable to outside stress is PSII [20,21]. Consistent with our experimental findings, it was demonstrated that UV-B stress not only decreased the photosynthetic parameters of chlorophyll fluorescence [22], but also considerably decreased Fv/Fo and Fv/Fm [23] (Figure 1). The current investigation demonstrated that UV-B radiation impacted light energy intake, transfer, and reaction to primary photochemistry in photosynthesis, resulting in leaf senescence, decreased chlorophyll content, carotenoid accumulation, and other effects (Figure 1 and Figure 2). In addition, UV-B rays harm PSII’s cyst-like membrane proteins, including the D1 and D2 proteins, which have an impact on plant photosynthesis. This is consistent with previous studies of cowhide cuckoos [4]. The onset of the K-phase in the kinetic curve of *R. chrysanthum* induced by chlorophyll fluorescence in response to UV-B stress suggested that the oxygen-evolving complex (OEC) damage on the donor side of PSII, the photoinhibition on the donor and acceptor side of PSII, the reduction in PSII stability, and the inactivation of the reaction centers were the factors that ultimately caused a decline in PSII’s overall performance (Figure 3 and Figure 4). However, through the photosynthetic system, UV-B irradiation can cause peroxidation of membrane lipids in plant tissues, which produces a lot of free radicals and upsets the balance of osmotic pressure within the cell. The photosynthetic system’s commencement of synthesis and accumulation of carotenoids may help plants regain their redox equilibrium and are crucial to the stability of photosynthesis.

The *R. chrysanthum*’s carotenoid and ABA contents were shown to be correlated by further analysis (Figure 2). This link would suggest that carotenoid and ABA biosynthesis are stimulated by UV-B-induced modifications in the photosynthetic system. We hypothesized that ABA could mitigate UV-B stress-induced damage to *R. chrysanthum*, or more accurately, that ABA helped *R. chrysanthum* withstand UV-B stress by mitigating damage to the donor-side of *R. chrysanthum*. Subsequent experiments confirmed that the K dots that developed after donor-side damage in *R. chrysanthum* after enduring UV-B stress vanished following external application of ABA (Figure 3e). This is consistent with the finding that under low temperature stress, abscisic acid mitigates alterations in carotenoids in maize seedlings and controls structural genes in the carotenoid biosynthesis pathway [24]. Conversely, we discovered that performance-parameter-characterizing PI_ABS_ had a moderating impact following ABA therapy, which is in line with our hypothesis (Figure 4d).

DEGs and DMs associated with carotenoid production were found in order to clarify the molecular mechanism of carotenoid accumulation in *R. chrysanthum*. In *R. chrysanthum*, ABA treatment up-regulated nearly every gene involved in carotenoid biosynthesis, including *dxr*, a member of the family that makes up the main rate-determining enzyme of the MEP pathway in the terpene skeleton biosynthesis pathway. We discovered that the results of tissue-specific investigations on *Artemisia* terpene biosynthesis agree with our own [25]. Consequently, *R. chrysanthum* showed increased accumulation of the carotenoid after treatment with ABA as opposed to UV-B stress treatment. Furthermore, following the application of ABA, *PDS* and *ZDS* showed higher expression levels. The process of converting octahydro lycopene (*C40*) into ζ-carotene is catalyzed by octahydro lycopene desaturase (*PDS*). Subsequently, ζ-carotene isomerizes into lycopene under the direction of ζ-carotene desaturase (*ZDS*) and carotenoid isomerase (*CRTISO*) [26]. As a primary branching point of the carotenoid biosynthesis pathway, lycopene plays a crucial function in the metabolic pathway of carotenoids. Cyclization with *LYC-b* or *LYC-e* and *LYC-b* allows it to generate α- or β-carotene.

The transcriptome data showed that, following ABA treatment, *R. chrysanthum* showed down-regulation of nearly all genes involved with photosynthesis and plant circadian rhythms, including *PIF3*, a gene that controls *CCA1*. Conversely, *CK2α* and *CK2β* are important modulators of plant circadian rhythms, and ABA treatment significantly reduced the expression of the CK2α-regulating gene, *CSNK2A*. This observation, however, defies prior research, and it could be explained by the *R. chrysanthum*’s distinct ABA-regulating mechanism [27]. The PSII light-harvesting chloroplast protein complex’s pertinent genes were found to be down-regulated. Additionally, we discovered that the OEC-regulating *psbP* genes had undergone significant alteration, indicating that PSII had been damaged by UV-B stress. These findings are consistent with the experimental data presented in Figure 1 and Figure 3.

By transcriptionally activating or repressing structural genes in the terpene biosynthesis pathway, TFs in plants control the metabolism of terpenes. According to earlier research, plants’ terpene metabolism may be regulated by members of the *AP2*, *bHLH*, *MYB*, *NAC*, *WRKY*, and *bZIP* TF families [25]. This study also found several of these transcription factors in a similar manner. Following the construction of nine-quadrant maps based on DEGs and DMs in the metabolic pathways, transcriptomics and metabolomics correlation analyses were carried out to identify four terpenoids that were significantly up-regulated (Adoxosidic acid, cichorioside B, Negundoin A*, and Vitexilactone*) and one terpenoid that was significantly down-regulated (Kankanoside E) following the application of ABA.

## 4. Materials and Methods

### 4.1. Plant Material, Growing Conditions, and Treatments

Nine *R. chrysanthum* seedlings under identical growing conditions were chosen, and each group was given three biological replications for the experimental treatments. Prior to being exposed to radiation, *R. chrysanthum* seedlings were raised in an intelligent artificial climate chamber that had the following settings: 18 °C/16 °C (day/night), 14 h/10 h (day/night), 50 µmol(photon) m^−2^s^−1^ of effective irradiance, and 60% relative humidity.

Two groups were cultured in normal 1/4 MS medium and received PAR radiation, the control (M), and PAR+UV-B radiation, UV-B stress-treated group(N). The other group was cultured in 1/4 MS medium supplemented with ABA (100 μmol/L) and received PAR+UV-B radiation, and was exogenously ABA-treated (Q). A 400 nm filter (Edmund, Filter Long 2IN SQ, Barrington, NJ, USA) was used to achieve PAR radiation on the culture flasks (effective irradiance of 50 µmol(photon) m^−2^s^−1^), and a 295 nm filter was used to achieve PAR+UV-B radiation on the same flasks (effective irradiance of 2.3 W/m^2^). For two days, eight hours each day of radiation therapy were administered concurrently to all three groups. Warm white fluorescent tubes (Philips, T5 × 14 W, Amsterdam, The Netherlands) produced visible light, while UV-B fluorescent tubes (Philips, Ultraviolet-B TL 20 W/01 RS, Amsterdam, The Netherlands) provided UV-B radiation. Based on research conducted in our lab, a modified version of the experiment was designed [28].

### 4.2. Chlorophyll and Carotenoid Content Determination

Concentrations of carotenoid and chlorophyll were measured using the method of Lichtenthaler and Buschmann [29], with the necessary modifications. After combining an aliquot (0.10 g) of the powdered material with 5 mL of 95% ethanol, the mixture was left to cure for 24 h in the dark. Using a spectrophotometer (Hitachi U-3000; Hitachi, Ltd., Chiyoda, Tokyo, Japan), chlorophyll extracts were filtered and examined. The following are the UV absorption wavelengths of total carotenoids, Chl a, and Chl b. The absorption peak of chlorophyll b is measured at 649 nm, total carotenoids at 470 nm, and chlorophyll an at 665 nm. The amounts of total carotenoids, also known as carotenoids, and the concentrations of Chl a (mg/g), Chl b (mg/g), and Chl a/b (mg/g) are also recorded.

### 4.3. OJIP Transient Measurement and Rapid Fluorescence Induction Kinetics Analysis

Measurements of fully grown leaves that were still attached were made for the Chlorophyll Fluorescence Transient (OJIP) profiles. A Handy-PEA (Handy plant efficiency analyzer, Hansatech Instruments Ltd., King’s Lynn, UK) was used for monitoring and analysis. Leaf clamps were averaged over three leaf detection points, and the leaf was dark-adapted for thirty minutes. Following dark adaption, the measurement holes were exposed to the laser light source by turning on the slide switch and attaching the instrument probe to the leaf clamps. To acquire fluorescence signals quickly, a 3000 μmol·m^−2^·s^−1^ preset LED light source with a 1 s detection time was employed. When all PSII reaction centers (RCs) were open (O step), the minimum fluorescence intensity signal (Fo) was measured at 20 μs, and when all RCs were closed (P step), the maximum intensity signal (Fm) was measured at 200–500 ms [30]. According to Kalaji et al. (2017) [31], the fluorescence intensities at 30 ms (I-step), 2 ms (J-step), and 300 μs (K-step) were designated as F_I_, F_J_, and F_K_.

In order to provide a thorough evaluation of the O-K and O-J cycles, we applied the following normalization to the raw transient profiles: W_O-K_ = (F_t_ − F_o_)/(F_K_ − F_o_) and W_O-J_ = (F_t_ − F_o_)/(F_J_ − F_o_) [32,33]. Using the equations ΔW_O-K_ = [W_O-K_ (stress) − W_O-K_ (control)], we examined the kinetic differences based on variable relative fluorescence to ascertain the presence of the K band [34]. The JIP test was used to calculate the OJIP transient parameters [35]. Table 2 displays the JIP test parameter equations along with their descriptions.

### 4.4. Establishing Parameters for Chlorophyll Fluorescence

The chlorophyll fluorescence parameter was determined using the IMAGING-PAM modulated chlorophyll fluorescence imaging system (HeinzWalz, Effeltrich, Germany). The leaves underwent a 30 min dark adaptation period before the test, during which the initial fluorescence Fo was measured and the maximum fluorescence Fm was generated by stimulation with saturating pulsed light. After the fluorescence was reduced to Fo, saturation pulses were turned on every 20 s for fluorescence measurements of the maximal fluorescence yield (Fm′) when the PS II reaction centers were all in the off state, and the actual fluorescence intensity (Fm). The photochemical (1600 μmol·m^−2^·s^−1^)-induced fluorescence kinetics were used. The final measured parameters include the photochemical fluorescence quenching coefficient qP, the nonphotochemical fluorescence quenching coefficient NPQ, and the highest photochemical efficiency of PSII Fv/Fm.

### 4.5. UPLC-MS/MS Investigation of R. chrysanthum Metabolites

Metware Biotechnology Co. Ltd. (Wuhan, China) isolated and analyzed all metabolites from the UV-B-treated group as well as exogenous ABA-treated *R. chrysanthum* histocultures seedlings. In a freeze-dryer (Scientz-100F, Hangzhou, China), leaf samples were vacuum freeze-dried, and after that, they were ground into a powder using a grinder (MM 400, Retsch, Haan, Germany). A solution of 1200 μL of 70% methanol by volume was used to dissolve 50 mg of powder. After extracting the homogenate for an entire night at 4 °C, it was centrifuged for three minutes at 4 °C at 12,000 rpm. After being filtered using a 0.22 μm microporous filter membrane, the resultant supernatant was placed in a sample vial for storage.

A Shimadzu Nexera X2 apparatus (Shimadzu, Kyoto, Japan) fitted with an Agilent SB-C18 column (1.8 μm, 2.1 × 100 mm, Santa Clara, CA, USA) was used to perform ultra performance liquid chromatography (UPLC). The mobile phase was made up of acetonitrile with 0.1% formic acid (solvent B) and ultrapure water with 0.1% formic acid (solvent A). In total, 0 min, 5%; 0–9 min, up to 95%; 9–10 min, 95%; 10–11.10 min, decreased to 5%; 11.10–14 min, 5% (17) was the gradient of solvent B. The injection volume was 4 μL, and the column temperature was set at 40 °C.

Applied Biosystems 4500 QTRAP equipment (ABI, Framingham, MA, USA) was used for tandem mass spectrometry (MS/MS) analysis. An API 4500 QTRAP UPLC-MS/MS system with an ESI turbo-ion spray interface was used to acquire a linear ion trap (LIT) and triple quadrupole (QQQ) scans. The operating conditions of the ESI source were as follows: the ion source was turbo-sprayed at 500 °C, and the ion-spray (IS) voltage was 5500 V for the positive ion mode and −4500 V for the negative ion mode. There were high ionization sensing parameters; Ion Source Gas I (GSI), Ion Source Gas II (GSII), and CUR set to 50, 60, and 25 psi, respectively; and negative ion mode, −4500 V [36].

Based on secondary spectrum data from the Metware Biotechnology Co.-compiled MWDB database, qualitative data were evaluated. Differentially accumulating compounds (DMs) were characterized as chemicals with variable significance in projection (VIP) > 1 and FC ≥ 1.2. The pathway enrichment study of these chemicals was conducted using the KEGG.

### 4.6. Building cDNA Libraries and Analyzing Transcriptomics Data

To process whole RNA, the mRNA enrichment approach was employed [37]. Using the Oligitex mRNA kit and Oligo (dT) magnetic beads, PolyA tail mRNA was extracted. Using the lysed mRNA as a template, the right amount of lysing agent was added at high temperatures to generate first-strand cDNA. The second-strand cDNA was subsequently created using a synthetic reaction system, and it was subsequently recovered, purified, and repaired with the aid of the kit. The “A” at the “3” end of the purified cDNA is joined to the mucus end.

The size of the altered fragments and the results of the final PCR amplification were utilized to calculate the product. Using an ABI StepOnePlus real-time PCR equipment and an Agilent 2100 Bioanalyzer, the quality of the library was evaluated. Reads with a quality value of less than 15 bases, which accounts for more than 50% of the total number of bases in that read, were classified as low-quality reads. Contaminants, reads with more than 10% unknown base N content, and low-quality reads were filtered and removed using SOAPnuke to guarantee data quality and accuracy. After aligning each sample’s gene expression levels to the reference gene sequences with the Bowtie2 (v2.25)3 program, the RESM (v1.2.8) software program was used to compute each sample’s gene expression levels.

A test to find DEGs in *R. chrysanthum* in response to UV-B exposure was conducted using the DEseq R program. The study found that DEGs for false discovery rates were Q-value (Adjusted *p*-value) ≤0.05.

### 4.7. Data Analysis

Every experiment mentioned above was carried out three times using a fully randomized design. Using SPSS (Version 21, SPSS Inc., Chicago, IL, USA), an analysis of variance (ANOVA) was performed on all of the data. The Duncan test was used to differentiate mean differences at the *p* < 0.05 level if significance was found. The standard deviation ± mean is used to report data.

## 5. Conclusions

This study’s findings showed that *R. chrysanthum*’s carotenoids were largely accumulated as a result of the plant’s response to UV-B stress. The up-regulation of *PDS* and *ZDS* may be the cause of *R. chrysanthum*’s increased carotenoid concentration during UV-B stress. *PDS* and *ZDS* are up-regulated when ABA is applied, and an increase in the expression of these genes raises the concentration of carotenoids. The rise in carotenoids corresponds with the rise in ABA levels. These results led to the proposal of a mechanism controlling the accumulation of cucurbitacin in the *R. chrysanthum*. UV-B radiation may cause systemic stress and light, which in turn stimulates the synthesis of ABA. This in turn stimulates the expression of transcription factors, like MYB, which in turn stimulates the expression of genes involved in the biosynthesis of carotenoids, improves their biosynthesis, and limits the build-up of chlorophyll. This work sheds important light on how the exogenous phytohormone ABA regulates carotenoid production in *R. chrysanthum* under UV-B exposure and suggests a possible method for enhancing seedling quality and UV tolerance.

## Figures and Tables

**Figure 1 plants-13-01062-f001:**
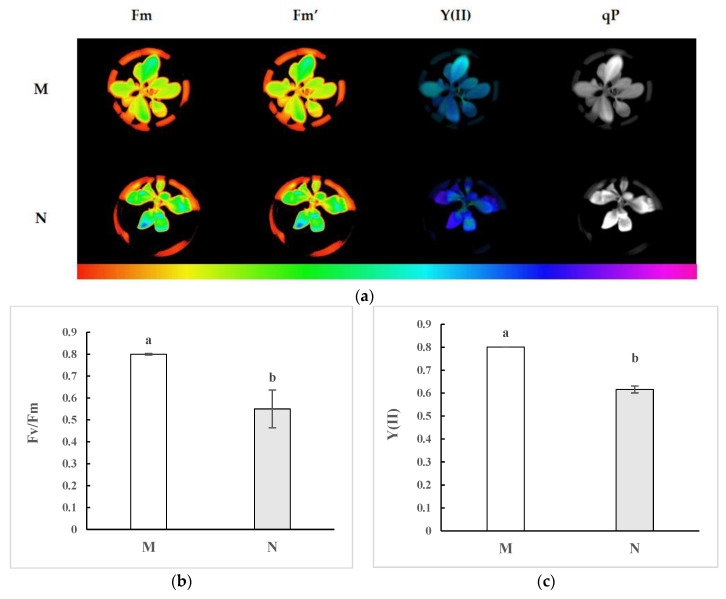
Changes in fluorescence parameters of the *R. chrysanthum* under UV-B stress. (**a**) Comparison of real-time fluorescence images of *R. chrysanthum* in control (M) and UV-B stress-treated group (N). (**b**–**e**) The bar graphs represent the maximum quantum yield of PSII (Fv/Fm), the actual quantum yield of PSII (Y(II)), the maximum fluorescence after photoacclimatization (Fm’), and the maximum photoconversion potential of PSII (Fv’/Fo’) in two groups of *R. chrysanthum*, the control group (M), and the UV-B stress-treated group (N), respectively. (**f**,**g**) The line graphs show the electron transport rate (ETR, μmol e^−1^ s^−1^ m^−2^) and NPQ as a function of PAR for the *R. chrysanthum* control (M) and UV-B stress-treated group (N), respectively. Values are means ± SD (*n* = 3). Different letters indicate significant difference at *p* < 0.05 among treatments.

**Figure 2 plants-13-01062-f002:**
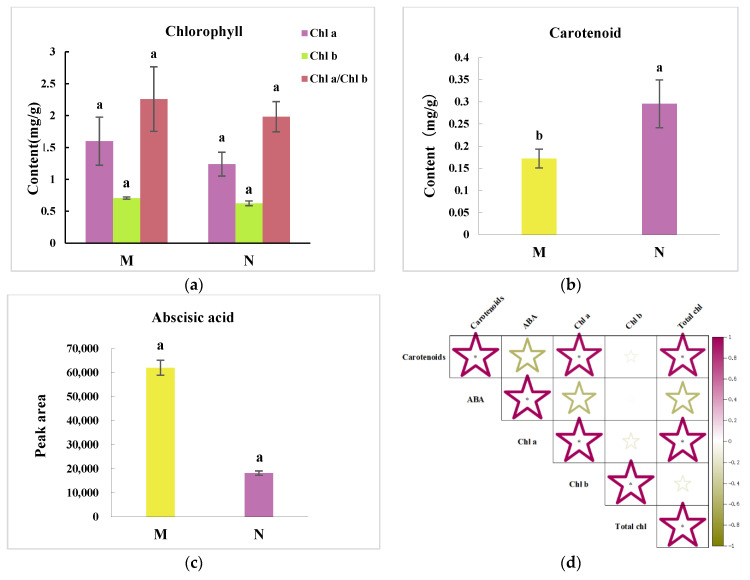
ABA is correlated with changes in *R. chrysanthum*’s carotenoid concentration following UV-B stress. (**a**) Variations in control (M) and UV-B stress-treated group (N) levels of Chl a, Chl b, and Chl a/Chl b. (**b**) Carotenoid content variations in control (M) and UV-B stress-treated group (N). (**c**) Changes in ABA abundance in Control (M) and UV-B stress-treated group (N). (**d**) Correlation study examining the relationship between variations in control (M) and UV-B stress-treated group (N). ABA abundance and carotenoid content. Green and purple star denote negative and positive correlations, respectively, and the size of the star reflects the correlation’s magnitude. Values are means ± SD (*n* = 3). Different letters (a, b) indicate significant difference at *p* < 0.05 among treatments. Significant difference (* *p* < 0.05).

**Figure 3 plants-13-01062-f003:**
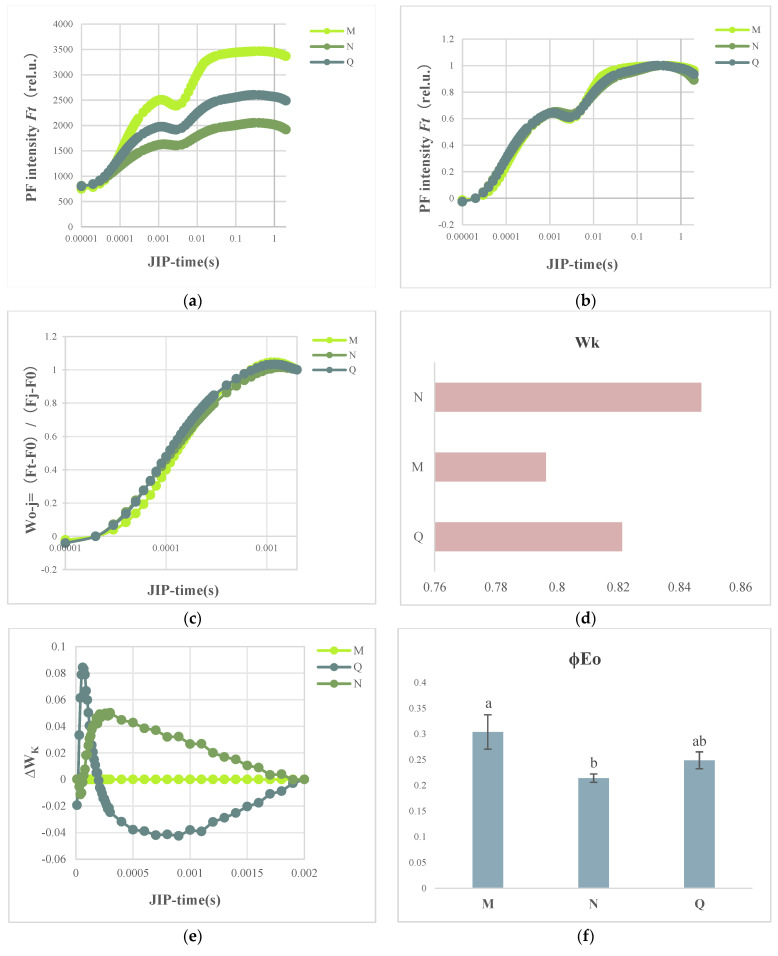
*R. chrysanthum* rapid fluorescence curves following UV-B stress and exogenous ABA application. (**a**) Shows the rapid fluorescence curves for the OP section. (**b**,**c**) Denotes the fast light curves of the OP section after normalization using W_O-P_ = (Ft − Fo)/(Fm − Fo) and the OJ section after normalization with W_O-J_ = (Ft − Fo)/(FJ − Fo), respectively. (**d**) W_K_ = (FK − Fo)/(FJ − Fo), which represents the ratio of the variable fluorescence *F*_K_ to the amplitude FJ − Fo. (**e**) Standardized fluorescence differences between the control(M), UV-B stress-treated(N), and exogenous ABA-treated (Q) OJs. (**f**,**g**,**h**) Stands for ϕEo, Sm, and ETo/RC variation, respectively. Values are means ± SD (*n* = 3). Different letters (a, b) indicate a significant difference at *p* < 0.05 among treatments.

**Figure 4 plants-13-01062-f004:**
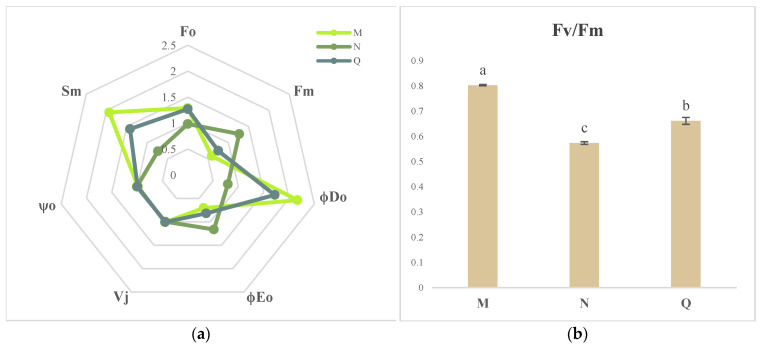
JIP parameters for control (M), UV-B stress treatment (N), and exogenous ABA treatment (Q) of *R. chrysanthum* were determined from the chlorophyll fluorescence OJIP transient curves. (**a**) Radargram showing how the values fluctuated in control (M), UV-B stress treatment (N), and exogenous ABA treatment (Q). (**b**–**d**) Indicate the variations in the PSII-related metrics PI _ABS_, DIo/RC, and Fv/Fm, respectively. Values are means ± SD (*n* = 3). Different letters (a, b, c) indicate significant difference at *p* < 0.05 among treatments.

**Figure 5 plants-13-01062-f005:**
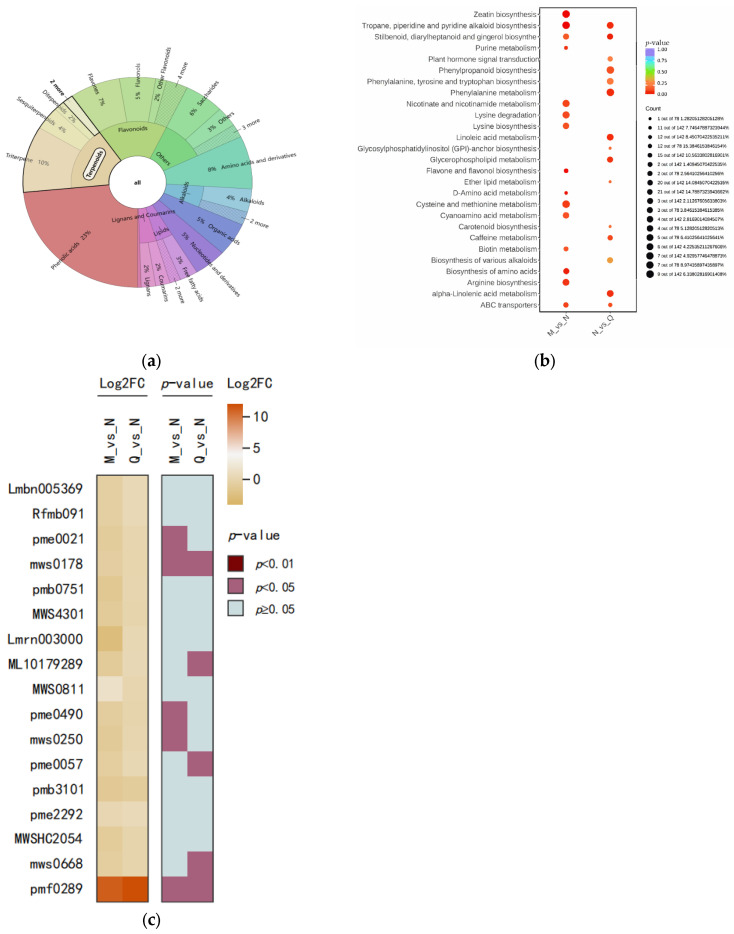
Enrichment pathways and occupancy of different metabolites in *R. chrysanthums* after UV-B stress and exogenous ABA treatment. (**a**) Distribution of distinct differential metabolites by percentage and primary and secondary classification. The circle near the center represents the primary classification of the metabolite and the outer circle represents the secondary classification of the metabolite. The lighter color highlights the terpenes. (**b**) Metabolite enrichment for each difference between the control (M) and UV-B stress-treated group (N) and between the UV-B stress-treated (N) and exogenous ABA-treated group (Q); larger circles indicate more metabolites enriched in the pathway, and redder colors indicate smaller and more significant *p*-values. (**c**) Heatmap of differential metabolite combinations in the *p* < 0.05 metabolic pathway: the right heatmap is calculated as a *p*-value, light blue indicates *p* ≥ 0.05, and pink indicates *p* < 0.05 with significant difference. The left heatmap is calculated as log_2_FC. Darker color indicates greater change in metabolite abundance, lighter color indicates lesser change in metabolite abundance.

**Figure 6 plants-13-01062-f006:**
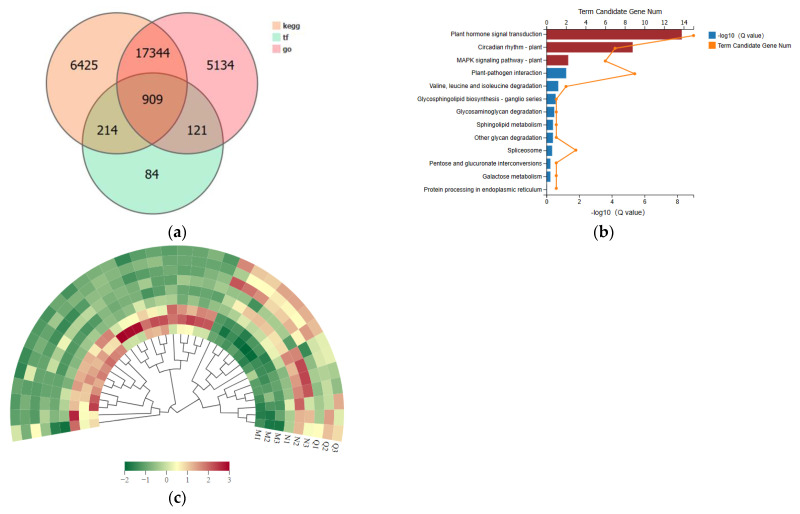
Following exogenous ABA treatment and UV- B stress, *R. chrysanthum* exhibits enrichment of important genes and transcription factor screening. (**a**) Annotation of key genes to Venn diagrams in KEGG, GO, and TF. (**b**) Selecting key genes for KEGG enrichment analysis through screening. Significantly enriched pathways (*p* < 0.01) are indicated by the three red-highlighted pathways. (**c**) The top five genes’ transcription factor families, ranked numerically, are analyzed for expression patterns. Redder colors indicate higher expression, while greener colors indicate lower expression.

**Figure 7 plants-13-01062-f007:**
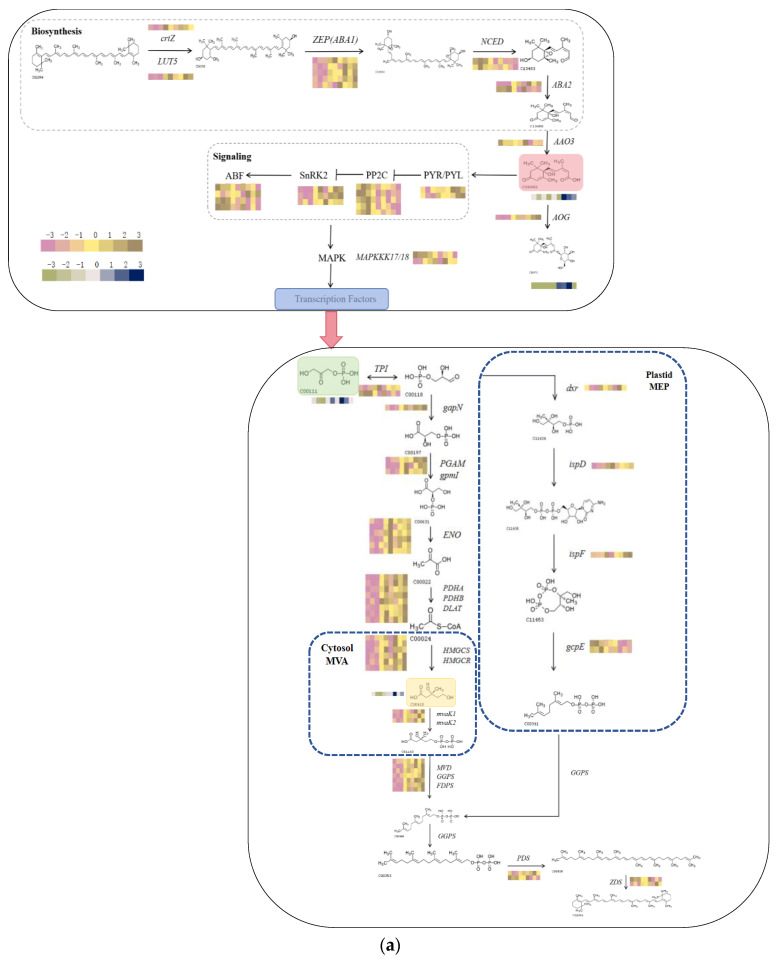
DEGs and DMs change following exogenous ABA therapy and UV-B stress in the *R. chrysanthum* carotenoid pathway. (**a**) ABA production, signal transduction pathways, and the biosynthesis of carotenoids. DEGs involved in the carotenoid biosynthesis pathway: *crtZ*, beta-carotene 3-hydroxylase; *LUT5*, beta-ring hydroxylase; *ZEP*, zeaxanthin epoxidase; *NCED*, 9-cis-epoxycarotenoid dioxygenase; *ABA2*, xanthoxin dehydrogenase; *AAO3*, abscisic-aldehyde oxidase; *PYR/PYL*, pyrabactin resistance–like; *PP2C*, protein phosphatase 2C; *SnRK2*, sucrose non–fermenting–1–related protein kinase 2; *ABF*, ABA-responsive element–binding factors; *AOG*, abscisate beta-glucosyltransferase; *MAPK*, mitogen–activated protein kinase; *TPI*, triosephosphate isomerase; gapN, glyceraldehyde-3-phosphate dehydrogenase; *gpmI*, 2,3-bisphosphoglycerate-independent phosphoglycerate mutase; *PGAM*, 2,3-bisphosphoglycerate-dependent phosphoglycerate mutase; ENO, enolase; *PDHA*, pyruvate dehydrogenase E1 component subunit alpha; *PDHB*, pyruvate dehydrogenase E1 component subunit beta; *DLAT*, pyruvate dehydrogenase E2 component (dihydrolipoyllysine-residue acetyltransferase); *HMGCS*, hydroxymethylglutaryl-CoA synthase; *HMGCR*, hydroxymethylglutaryl-CoA reductase; *MVK*, mevalonate kinase; *MVD*, diphosphomevalonate decarboxylase; *GGPS*, geranylgeranyl diphosphate synthase, type II; *FDPS*, farnesyl diphosphate synthase; *dxr*, 1-deoxy-D-xylulose-5-phosphate reductoisomerase; *ispD*, 2-C-methyl-D-erythritol 4-phosphate cytidylyltransferase; *ispF*, 2-C-methyl-D-erythritol 2,4-cyclodiphosphate synthase; *gcpE*, (E)-4-hydroxy-3-methylbut-2-enyl-diphosphate synthase; *PDS*, 15-cis-phytoene desaturase and *ZDS*, zeta-carotene desaturase. DMs involved in the carotenoid biosynthesis pathway: C06082, Abscisate; C15970, Abscisic acid glucose ester; C00111, Glycerone phosphate and C00416, Phosphatidate. (**b**) An overview of photosynthetic processes and circadian rhythms in plants. DEGs involved in photosynthesis and circadian pathways in plants: *PIF3*, phytochrome-interacting factor 3; *CSNK2A*, casein kinase II subunit alpha; *LHCA2*, light-harvesting complex I chlorophyll a/b binding protein 2; *LHCA3*, light-harvesting complex I chlorophyll a/b binding protein 3; *LHCA5*, light-harvesting complex I chlorophyll a/b binding protein 5; *PsbP*, photosystem II oxygen-evolving enhancer protein 2; *PetH*, ferredoxin--NADP+ reductase and *petJ*, cytochrome c6. Where heat maps depict changes in genes and metabolites. On the metabolite heat map, a significant down-regulation is indicated by a green hue, while significant up-regulation is indicated by a bluer color. Considerable down-regulation is shown by the gene heatmap’s more purple color, while considerable up-regulation is indicated by its more brown color.

**Figure 8 plants-13-01062-f008:**
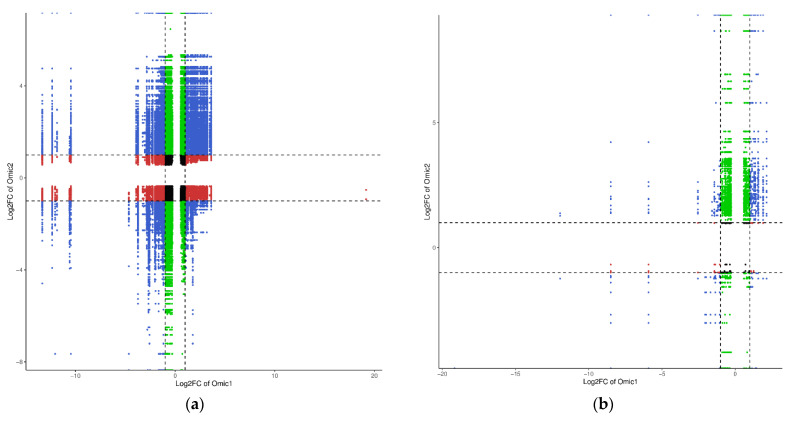
Important genes and terpenoids in *R. chrysanthum* following UV-B stress and exogenous ABA injection during UV-B stress. (**a**,**b**) Nine-quadrant plots of the DEGs and DMs following external application of ABA under UV-B stress and UV-B stress, respectively, where the log_2_FC of DMs is represented by the horizontal coordinates, and the log_2_FC of DEGs is represented by the vertical coordinates. Quadrants 1 and 9 are characterized by a positive correlation between the horizontal and vertical coordinates; quadrants 3 and 7 are characterized by a negative correlation between the horizontal and vertical coordinates; quadrant 5 is characterized by no correlation between the horizontal and vertical coordinates; quadrants 2 and 8 are characterized by a significant change in the horizontal coordinates without a significant change in the vertical coordinates; quadrants 4 and 6 are characterized by a significant change in the vertical coordinates without a significant change in the horizontal coordinates. (**c**,**d**) Number of DMs and DEGs in each quadrant for the control group (M) versus the UV-B stress-treated group (N) and the UV-B stress-treated group (N) versus the exogenous ABA-treated group (Q). (**e**,**f**) Chord plots of DEGs and key terpene correlations in quadrants 3 and 7, respectively. A stronger positive correlation is shown by a redder color, and a stronger negative correlation is indicated by a bluer tint.

**Table 1 plants-13-01062-t001:** Top 15 genes’ information displayed in the sorted table by *p* ≤ 0.05.

Gene ID	Q-Value (M-vs.-N)	log_2_ (N/M)	Q-Value (N-vs.-Q)	log_2_ (Q/N)
TRINITY_DN10705_c0_g2_i1-A1	0.035302	1.195234	0.998232	−0.007671
TRINITY_DN12335_c0_g4_i1-A1	0.025145	2.002006	0.997827	0.684958
TRINITY_DN154_c0_g5_i1-A1	0.023931	−1.070521	0.997827	−0.686958
TRINITY_DN17958_c0_g2_i1-A1	0.037824	−1.997859	0.997827	−1.220038
TRINITY_DN20110_c0_g1_i1-A1	0.024333	0.685827	0.997827	−0.274792
TRINITY_DN20522_c0_g1_i1-A1	0.000908	−3.206042	0.997827	−0.404955
TRINITY_DN21509_c0_g1_i1-A1	0.009498	−1.769117	0.127622	−2.776231
TRINITY_DN2213_c0_g1_i4-A1	0.010107	−0.844769	0.997827	−0.216735
TRINITY_DN236_c0_g1_i2-A1	0.046394	0.944447	0.997827	−0.442235
TRINITY_DN236_c0_g1_i3-A1	0.001978	1.461774	0.997827	−0.601384
TRINITY_DN236_c0_g1_i6-A1	0.002181	1.409949	0.997827	−0.364035
TRINITY_DN2428_c0_g1_i1-A1	0.006753	−0.980128	0.997827	−0.313276
TRINITY_DN249_c0_g1_i5-A1	0.000000	−1.777728	0.997827	0.321110
TRINITY_DN27405_c0_g1_i1-A1	0.000006	−2.997685	0.997827	0.766613
TRINITY_DN2967_c1_g1_i1-A1	0.004706	0.979329	0.413874	1.099604

Notes: Group M is the control treatment, group N is the UV-B stress treatment, and group Q is the exogenous ABA treatment group.

**Table 2 plants-13-01062-t002:** Equations and summaries of parameters taken from transient data on chlorophyll fluorescence (OJIP) were applied.

Parameters	Explanation
Parameters derived from OJIP transient data	
Fo = F_20μs_	Minimum fluorescence
Fk = F_300μs_	300 μs instantaneous fluorescence
FJ = F_2ms_	2 ms instantaneous fluorescence
Fi = F_30ms_	30 ms instantaneous fluorescence
Fp = Fm	Maximum fluorescence
Fv/Fm = (Ft − Fo)/(Fm − Fo)	The PSII’s maximum photochemical efficiency
V_J_ = (FJ − Fo)/(Fm − Fo)	Relative variable fluorescence intensity at the J-step
V_I_ = (FI − Fo)/(Fm − Fo)	Relative variable fluorescence intensity at the I-step
V_K_ = (FK − Fo)/(Fm − Fo)	Relative variable fluorescence intensity at the K-step
W_K_ = (FK − Fo)/(FJ − Fo)	Ratio of variable fluorescence Fk to the amplitude FJ − Fo
Sm = (Area)/(Fm − Fo)	Normalized total complementary area
**Flux ratios of PSII**	
φEo = ETo/ABS = [1 − (Fo/Fm)] ψo	Quantum yield for electron transport (at t = Fo_)_
φDo = 1 − φpo = (Fo/Fm)	Quantum yield at t = Fo for energy dissipation
ψo = ETo/TRo = (1 − V_J_)	Probability that a trapped exciton moves an electron into the electron transport chain beyond QA-(at t = Fo)
**Activities per reaction center (RC)**	
DIo/RC = ABS/RC − TRo/RC	Dissipated energy flux per RC (at t = Fo)
ETo/RC = Mo(1/V_J_)/(1/V_J_)	Electron transport flux per RC (at t = Fo)
**Vitality indexes**	
PI_ABS_ = RC/ABS[φpo/1 − φpo)][ψo/(1 − ψo)]	Performance index on absorption basis

## Data Availability

The datasets generated during and/or analyzed during the current study are available from the corresponding author on reasonable request.

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
