# Peer review of "Carotenoid Accumulation in the Rhododendron chrysanthum Is Mediated by Abscisic Acid Production Driven by UV-B Stress"

_plants, 2024, doi:10.3390/plants13081062_

Round 1

Reviewer 1 Report

Comments and Suggestions for Authors

- Please explain the abbreviations appearing for the first time in the text, both in the abstract and in the body of the text.

- Please provide the specific UV-B radiation characteristics used in the study.

- Photographs of individual plant groups in white light must be provided

- The experiment is not clearly described e.g. „Three treatment groups (Groups M, N, and Q) were conducted: PAR, UV-B, and ABA + PAR” it follows from this sentence that M – PAR; N – UV-B; Q – ABA+PAR. However, the next sentence describes otherwise „(...) and plants of groups N and Q were subjected to UV-B radiation ( 2.3 W/m2). The three groups were subjected to 8 h/d of UV-B radiation for 2 d, and all the parameters were measured after 2 d of treatment.”

- In this case, I suggest preparing a methodological abstract.

- The naming of the groups is also unclear; it would be better to call the contorl group simply control

- „The OJIP (oxygen-juicipurine-photoluminescence)” – unclear

- Unclear description of the methodology and design of the experiment makes it extremely difficult to read the results correctly. Therefore, in addition to correcting the description of the methods, I suggest accurately captioning the figures and tables, e.g. in the caption of Fig. 1, Fig 2 it is necessary to clarify what the names of the groups M and N mean. Again I suggest to use „control” insted M

- How was the ABA analysed? Why is only the peak area given? Did you use a standard?

- „Metware Biotechnology Co. Ltd. isolated and analyzed the metabolites” – what kind of metabolites?

- Figure 5, 6, 7,8 – The quality of this figure is poor - the footnote needs to be increased.

Reviewer 2 Report

Comments and Suggestions for Authors

General comments:

The purpose and objectives of the study titled 'The accumulation of carotenoids in Rhododendron chrysanthum is mediated by the production of abscisic acid driven by UV-B stress' are clearly and concisely stated. The study proposes a methodology for investigating the accumulation of carotenoids in the cowpea azalea under UV-B radiation stress and the role of abscisic acid (ABA) in this process.

As R. crysanthum is a plant with high levels of carotenoid pigments, which provide antioxidant benefits and protect against various stresses, including radiation, it makes a suitable subject for studying the photosynthetic and metabolic mechanisms involved in this dichotomy.

The bibliography is up-to-date and not too dense. The background information is clearly presented and sufficient for understanding and monitoring the work.

The experimentation carried out to achieve the proposed objectives is appropriate and based on both the results and the discussion. It is a manuscript, easily to follow, which will make it more accessible to the scientific community related to the topic.

The present work will contribute to the main study of the accumulation of carotenoids under stress conditions and provide new tools to elucidate the intrinsic metabolic mechanisms of the appearance of said secondary metabolites. These mechanisms could also be extrapolated to other photosynthetic organisms.

Both the Results and Discussion sections are well-argued, developed, and clarified, providing scientific soundness and coherence.

The Materials and Methods section is also well-developed and described.

Additionally, the metabolomics and transcriptomics studies make a valuable contribution in terms of quality, complexity, and relevance, which enhances the reliability of the results and initial hypotheses.

Specific comments:

Abstract: the abstract, it is well-explained and presented. However, there are acronyms such as DEGs, DAFs, and OJIP that are not defined. Although these terms may be routine in the community dedicated to the subject, it is important to clarify them for readers who may not be familiar with them. The meaning of ABA has been indicated, so it is important to clarify the other acronyms as well. In reference to this, in the introduction for example, the clarification of "abscisic acid (ABA) is repeated (lines are not detailed in the manuscript so it is difficult to indicate exactly where).

Results: To improve clarity and facilitate follow-up of the manuscript, it would be recommended that the meaning of N groups, M groups and NM groups be clearly stated, as this information can be found in the Materials and Methods section following the Results section, it would be recommended that these meanings be clearly stated the first time they appear in the Results section, first time in the manuscript. Likewise, this should be described in the figure captions where they appear, even if they are already stated in the text of the manuscript, for example Figure 1.

In 2.1. section, PSII sometimes appears as PSII and other times as PS II, unifying formats throughout the manuscript (also in page 19).

R. chrysanthum should be always written in italics, example in pages 6, 10

Round 2

Reviewer 1 Report

Comments and Suggestions for Authors

The paper was significantly improved and can be published in its current form.